# TORC1 modulation in adipose tissue is required for organismal adaptation to hypoxia in *Drosophila*

Byoungchun Lee [1], Elizabeth C. Barretto[1] & Savraj S. Grewal[1]

Animals often develop in environments where conditions such as food, oxygen and temperature fluctuate. The ability to adapt their metabolism to these fluctuations is important for normal development and viability. In most animals, low oxygen (hypoxia) is deleterious. However some animals can alter their physiology to tolerate hypoxia. Here we show that TORC1 modulation in adipose tissue is required for organismal adaptation to hypoxia in *Drosophila*. We find that hypoxia rapidly suppresses TORC1 signaling in *Drosophila* larvae via TSC-mediated inhibition of Rheb. We show that this hypoxia-mediated inhibition of TORC1 specifically in the larval fat body is essential for viability. Moreover, we find that these effects of TORC1 inhibition on hypoxia tolerance are mediated through remodeling of fat body lipid storage. These studies identify the larval adipose tissue as a key hypoxia-sensing tissue that coordinates whole-body development and survival to changes in environmental oxygen by modulating TORC1 and lipid metabolism.

---

[1] Clark H Smith Brain Tumour Centre, Arnie Charbonneau Cancer Institute, Alberta Children's Hospital Research Institute, and Department of Biochemistry and Molecular Biology Calgary, University of Calgary, Calgary T2N4N1 Alberta, Canada. Correspondence and requests for materials should be addressed to S.S.G. (email: grewalss@ucalgary.ca)

Animals often have to grow and survive in conditions where their environment fluctuates. For example, changes in nutrition, temperature or oxygen availability, or exposure to pathogens can all impact development. Animals must therefore adapt their physiology and metabolism in response to these environmental challenges in order to ensure proper homeostasis[1,2].

In most animals decreases in oxygen are particularly deleterious. Low oxygen (hypoxia) can lead to rapid tissue damage and lethality, and oxygen deprivation is a hallmark of diseases such as stroke and ischemia[3]. However, some animals have evolved to live in oxygen-deprived conditions and consequently exhibit marked tolerance to hypoxia. For example, birds and aquatic mammals can tolerate extensive periods of low oxygen without incurring any tissue damage[4,5]. Indeed, some animals show quite remarkable levels of tolerance to oxygen deprivation: brine shrimp embryos have been reported to recover from four years of continuous anoxia[6], while the naked mole rat can survive up to 18 min of complete oxygen deprivation, a condition that kills laboratory rodents within about one minute[7]. Understanding how animals like these adapt their metabolism to low oxygen may shed light on how to protect tissues from hypoxic damage in disease states.

*Drosophila* provide an excellent laboratory model system to examine how changing environmental conditions influence animal development. In particular, there has been extensive work on how nutrient availability influences *Drosophila* larval development, the main growth period of the life cycle[8–10]. In nutrient-rich conditions larvae increase in mass ~200-fold over 4 days before undergoing metamorphosis to the pupal stage[11,12]. In contrast, when dietary nutrients are limiting, larvae alter their physiology and metabolism to slow growth and development, and to promote survival. One main regulator of these nutrient-regulated processes in *Drosophila* is the conserved TOR kinase signalling pathway[13]. TOR exists in two signalling complexes, TORC1 and TORC2, with TORC1 being the main growth regulatory TOR complex[14]. A conserved signalling network couples nutrient availability to the activation of TORC1 to control anabolic processes important for cell growth and proliferation[14]. Moreover, studies in *Drosophila* have been instrumental in revealing non-autonomous effects of TORC1 signalling on body growth. For example, nutrient activation of TORC1 in specific larval tissues such as the fat body, muscle and prothoracic gland, can influence whole animal development through the control of endocrine signalling via insulin-like peptides and the steroid hormone, ecdysone[9,10,15]. In addition, TORC1 regulation of autophagy in the larval fat body is important for organismal homeostasis and survival during periods of nutrient deprivation[16,17].

*Drosophila* larvae are also hypoxia tolerant[18–20]. In their natural ecology, *Drosophila* larvae grow on rotting food rich in microorganisms, which probably contribute to a low oxygen local environment. Even in the laboratory, local oxygen levels are low at the food surface of vials containing developing larvae[19]. *Drosophila* have therefore evolved metabolic and physiological mechanisms to tolerate hypoxia. However, compared to our understanding of the nutrient regulation of growth and homeostasis, less is known about how *Drosophila* adapt to low oxygen. A handful of studies have shown that larval survival in oxygen requires regulation of gene expression by the transcription factors HIF-1 alpha and ERR alpha, and the repressor, Hairy[21–24]. Developmental hypoxia sensing and signalling has also been shown to be mediated through a nitric oxide/cGMP/PKG signalling pathway[25,26].

Here we report a role for modulation of the TORC1 kinase signalling pathway as a regulator of hypoxia tolerance during *Drosophila* development. In particular, we find that suppression of TORC1 specifically in the larval fat body is required for animals to reset their growth and developmental rate in hypoxia, and to allow viable development to the adult stage. We further show that these effects of TORC1 inhibition require remodelling of lipid droplets and lipid storage. Our findings implicate the larval fat body as a key hypoxia-sensing tissue that coordinates whole animal development and survival in response to changing oxygen levels.

## Results

**Hypoxia slows larval growth and delays development**. We began by examining the effect of hypoxia on larval development. We used 5% oxygen as our hypoxia condition for all experiments in this paper. We allowed embryos to develop in normoxia and then, upon hatching, larvae were maintained on food in either normoxia or hypoxia. We found that hypoxia led to a reduced larval growth rate and larvae took approximately an extra 2 days to develop to the pupal stage (Fig. 1a). We also found that the hypoxia-exposed animals had a reduced wandering third instar larval weight (Fig. 1b) and reduced final pupal size (Fig. 1c). Exposure to hypoxia did not alter larval feeding behaviour (Supplementary Fig. 1a,b), suggesting that the decreased growth rate was not simply due to a general reduction in nutrient intake. These data indicate that *Drosophila* larvae adapt to low oxygen levels by reducing their growth and slowing their development. These data are consistent with previous reports showing that moderate levels of hypoxia (10% oxygen) can also affect final body size[20].

**Hypoxia suppresses TORC1 signalling via TSC1/2**. The conserved TORC1 signalling pathway is one of the main regulators of growth in *Drosophila*. TORC1 can be activated by dietary nutrients and growth factors such as insulin. Mammalian cell culture experiments have also shown that hypoxia can suppress TORC1 activity[27–30]. We therefore examined whether changes in TORC1 signalling play a role in larval hypoxia tolerance. We transferred third instar larvae from normoxia to hypoxia and then measured TORC1 activity by western blotting using an antibody that recognizes the phosphorylated form of S6 kinase (pS6K), a direct TORC1 kinase target. We found that hypoxia led to a rapid suppression of whole-body TORC1 activity that was apparent within 10–20 min of hypoxia exposure (Fig. 2a). This suppression persisted when larvae were maintained in hypoxia for longer periods (48 h, Supplementary Fig. 2a). We also examined how different levels of oxygen affected TORC1 activity. Third instar larvae were transferred from normoxia to different levels of hypoxia (from 20–1% oxygen) for 1 h and then TORC1 activity measured. We found that suppression of TORC1 occurred at 5 and 3% oxygen but remained unchanged at higher (20 and 10%) or lower (1%) levels (Fig. 2b, Supplementary Fig. 2b). We examined this further by exposing larvae to several different concentrations of oxygen between 1 and 10%, and found that the range within which TORC1 was inhibited was between 2 and 6 % oxygen (Supplementary Fig. 2C). These data indicate that larvae rapidly respond to hypoxia by suppressing TORC1 signalling, and that this response occurs within a specific range of low environmental oxygen rather than simply being triggered below a threshold level of low oxygen.

We next examined how hypoxia suppresses TORC1 activity. One of the main ways by which TORC1 is activated is through a TSC1/2-Rheb signalling pathway[14]. Rheb is a small G-protein that binds to and activates TOR kinase at lysosomes. TSC2 is a GTPase activating protein, and when bound to its partner TSC1, it inhibits Rheb by converting it from its active GTP-bound state

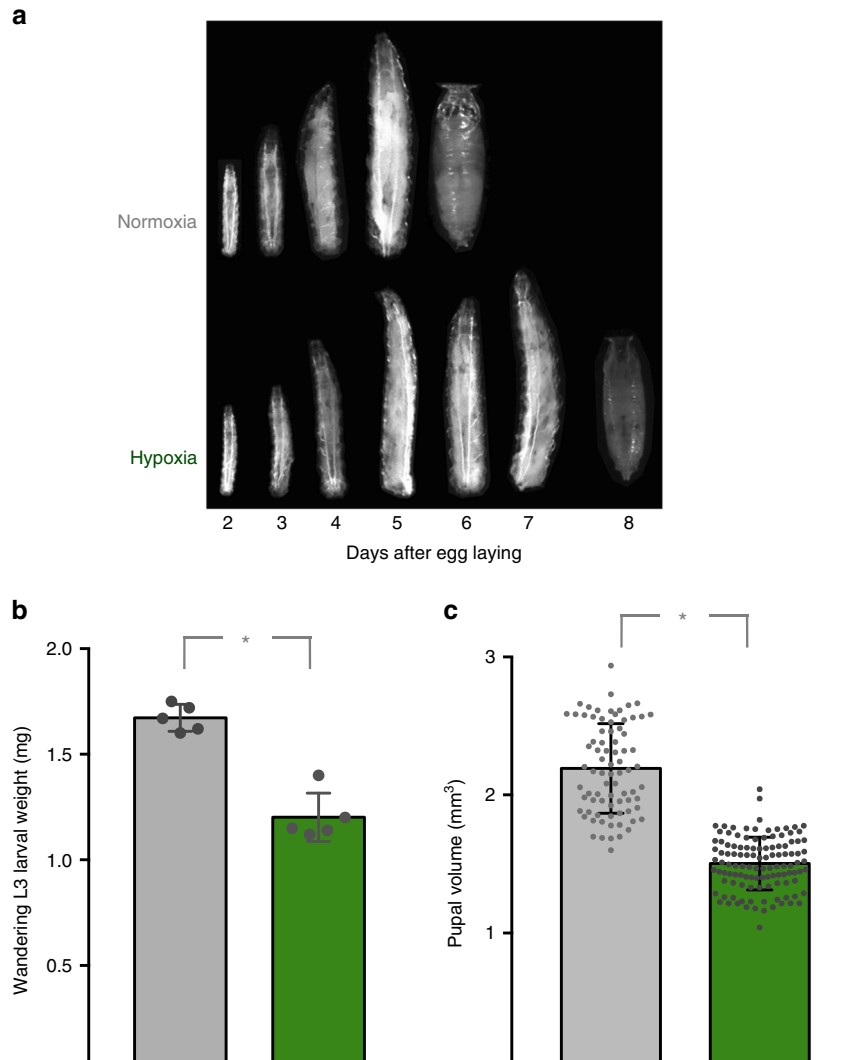

**Fig. 1** Hypoxia inhibits larval growth and development. **a** Larvae were hatched in normoxia and then either maintained in normoxia (top images) or transferred to hypoxia (5% oxygen, bottom images). Larvae and pupae were then subsequently imaged on each day following egg hatching. Hypoxia led to a delay in larval growth and development. **b** Larvae were hatched in normoxia and then maintained in either normoxia or hypoxia (5% oxygen) until the wandering third instar stage. Larval weights were then measured. Hypoxia led to a reduction in larval mass. Data are expressed as mean ± SEM. *$p < 0.05$, Students $t$-test. **c** Larvae were hatched in normoxia and then maintained in either normoxia or hypoxia (5% oxygen) until pupation. Pupal size was then measured. Hypoxia lead to a reduction final pupal size. Data are expressed as mean ± SEM, *$p < 0.05$, Students $t$-test

to an inactive GDP-bound state. Several diverse stimuli including nutrients, growth factors and hypoxia have been shown to regulate TSC1/2 function and to control TORC1 activity in mammalian cell culture[14]. We therefore explored a role for TSC1/2 and Rheb in the suppression of TORC1 kinase signalling during larval hypoxia. We found that ubiquitous overexpression of a *UAS-Rheb* transgene (using daughterless-gal4, *da-gal4*) prevented the hypoxia-mediated suppression of TORC1 signalling in larvae (Fig. 2c). We next tested a role for TSC1/2 function by examining *tsc1* mutants. We found that larvae that were either homozygous for a null mutation of *tsc1* (*tsc1$^{W240X}$*) or trans-heterozygous for two independent mutations (*tsc1$^{Q87X}$/tsc1$^{W240X}$*) were unable to suppress TORC1 signalling when exposed to hypoxia (Fig. 2d, Supplementary Fig. 3c). Together these data suggest that hypoxia inhibits TORC1 by TSC1/2-mediated suppression of Rheb.

Studies in mammalian cells have described how hypoxia can induce TSC-mediated TORC1 inhibition via the classic HIF-1 alpha transcription factor. In this mechanism, HIF-1 alpha leads to upregulation of REDD1, an activator of TSC1/2[31]. In

*Drosophila*, the homolog of REDD1, Scylla, and its paralog, Charybdis, have been shown to inhibit TOR and suppress growth[32]. We therefore examined a role for Sima (the *Drosophila* HIF-1 alpha homolog) and Scylla/Charybdis in larval hypoxia. We first tested a previously described sima mutant (*sima$^{07607}$*). Homozygous animals showed no sima expression (Supplementary Fig. 4a) and exhibited 100% lethality when maintained in hypoxia throughout the larval period as previously described (Supplementary Fig. 4b)[21,24]. However, we found that third instar *sima$^{07607}$* mutant larvae still showed a suppression of TORC1 signalling when transferred from normoxia to hypoxia (Fig. 2e). We also examined the effects of sima knockdown using RNAi. As expected, we found that ubiquitous overexpression of a *UAS-sima* inverted repeat (IR) dsRNA transgene (using *da-gal4*) led to lethality in hypoxia (Supplementary Fig. 4c) and also reduced expression of both *sima* mRNA and expression of a sima target gene, *fatiga* (Supplementary Fig. 4d). However, as with the *sima* mutants, *da > sima IR* animals still showed suppression of TORC1 signalling when exposed to hypoxia (Supplementary

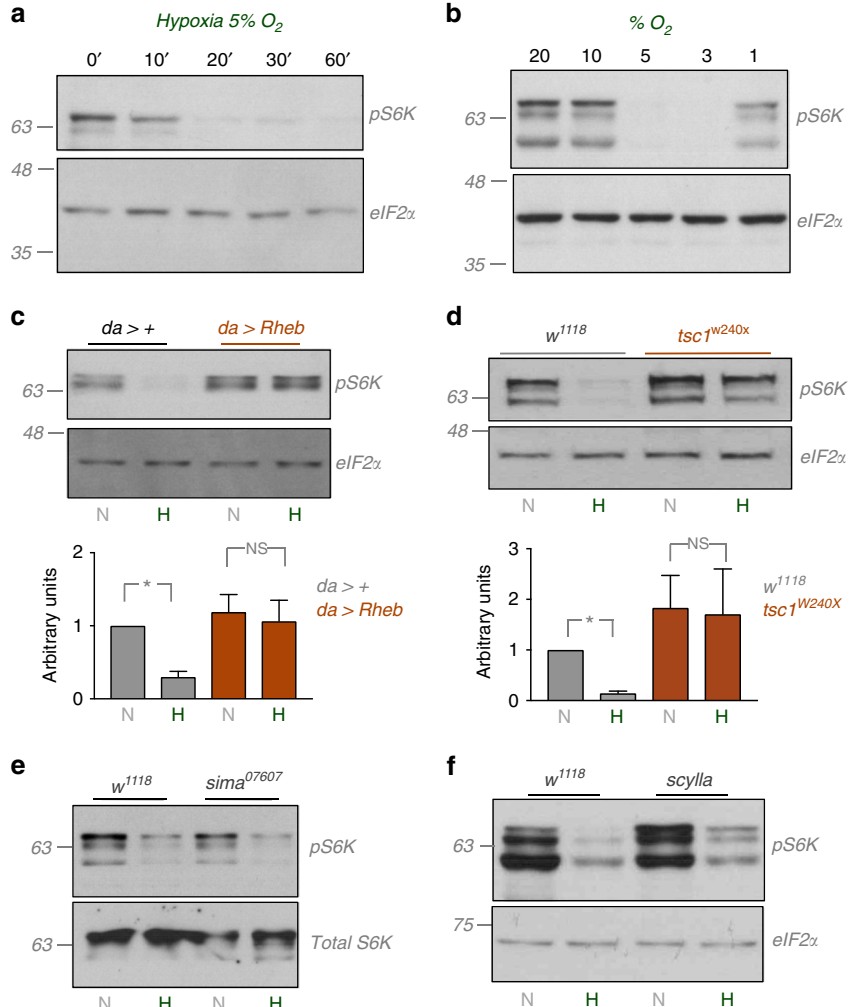

**Fig. 2** Hypoxia suppresses TORC1 signalling via TSC1/2. **a** Early third instar larvae were transferred from normoxia to hypoxia (5% oxygen). At the indicated times, larvae were then collected, lysed and processed for SDS-PAGE and western blotting using antibodies to phospho-S6K (pS6K) or total eIF2alpha (eIF2α). **b** Early third instar larvae were transferred from normoxia to different levels of hypoxia (20–1% oxygen) for 1 h. Larvae were then collected, lysed and processed for SDS-PAGE and western blotting using antibodies to phospho-S6K (pS6K) or total eIF2alpha (eIF2α). **c** Control (da>+) or Rheb overexpressing (da > Rheb) early third instar larvae were either maintained in normoxia (N) or transferred from normoxia to hypoxia (5% oxygen, H) for 1 h. Larvae were then collected, lysed and processed for SDS-PAGE and western blotting using antibodies to phospho-S6K (pS6K) or total eIF2alpha (eIF2α). Quantified band intensities from three independent experiments are shown below the blot. Data represent relative pS6K band intensities corrected for eIF2α (loading control) band intensity. Quantifications were performed using Image J. Data represent mean ± SD. *$p < 0.05$, Students $t$-test. **d** Control (w[1118]) or tsc1 mutant (tsc1[W240X]) larvae were either maintained in normoxia (N) or transferred from normoxia to hypoxia (5% oxygen, H) for 1 h. Larvae were then collected, lysed and processed for SDS-PAGE and western blotting using antibodies to phospho-S6K (pS6K) or total eIF2alpha (eIF2α). Quantified band intensities from three independent experiments are shown below the blot. Data represent relative pS6K band intensities corrected for eIF2α (loading control) band intensity. Quantifications were performed using Image J. Data represent mean ± SD. *$p < 0.05$, Students $t$-test. **e** Control (w[1118]) or sima mutant (sima[07607]) larvae were either maintained in normoxia (N) or transferred from normoxia to hypoxia (5% oxygen, H) for 1 h. Larvae were then collected, lysed and processed for SDS-PAGE and western blotting using antibodies to phospho-S6K (pS6K) or total S6K. **f** Control (w[1118]) or scylla mutant (scylla) larvae were either maintained in normoxia (N) or transferred from normoxia to hypoxia (5% oxygen, H) for 1 h. Larvae were then collected, lysed and processed for SDS-PAGE and western blotting using antibodies to phospho-S6K (pS6K) or total eIF2alpha (eIF2α)

Fig. 5b). Finally, we tested a role for Scylla and Charybdis. We found that larvae that were either homozygous mutant for *scylla* or *charybdis* or double-mutant for both genes showed the same suppression of TORC1 signalling as seen in wild-type control larvae (Fig. 2f, Supplementary Fig. 6).

We also explored a role for three other potential hypoxic regulators of TORC1: AMPK, which is activated under hypoxia in mammalian cell culture and can suppress TORC1 signalling, in part by phosphorylating and inhibiting TSC2[28,33,34]; Ptp61F, a phosphatase previously shown to regulate TORC1 in *Drosophila* cultured cells exposed to hypoxia[35]; and nitric oxide, which is

induced upon hypoxia in *Drosophila* and which is required for survival in low oxygen[25,26]. However, when we used RNAi to knockdown expression of AMPK alpha, ptp61F or nitric oxide synthase in larvae (using *da-GAL4* to drive ubiquitous expression of UAS RNAi transgenes), we saw that hypoxia exposure still led to a strong inhibition of TORC1 comparable to that seen in control animals (Supplementary Fig. 7). Together, our data suggest that the rapid suppression of TORC1 signalling upon hypoxia exposure in larvae requires TSC1/2 function but is independent of both HIF-1 alpha mediated transcription and the signaling molecules AMPK, Ptp61F and nitric oxide.

**Reduced TORC1 in the fat body is needed for hypoxia tolerance**. We next examined whether the suppression of overall TORC1 activity that we observed was important for tolerance to hypoxia during *Drosophila* development. We began by testing the effects of ubiquitous expression of *UAS-Rheb* with *da-Gal4*, since we found that in this condition larvae are unable to suppress TORC1 activity in hypoxia (Fig. 2c). We first compared development in control (*da>+*) vs. Rheb overexpressing (*da > Rheb*) animals that were grown throughout their larval period from hatching in either normoxia or hypoxia. We found that larval

Rheb overexpression had no effect on overall survival to the pupal stage in either normoxia or hypoxia (Fig. 3a). We next examined developmental rate by measuring the time to pupation. In normoxia, we found that Rheb overexpression (*da > Rheb*) led to a slight increase in developmental rate compared to control animals (*da>+*, Fig. 3b). When raised in hypoxia, the *da>+* animals had an approximately 2-day delay to pupation, and this developmental delay was even further exacerbated in *da > Rheb* animals (Fig. 3b). We also measured effects on overall body size. We found that *da > Rheb* animals exhibited an increase in both

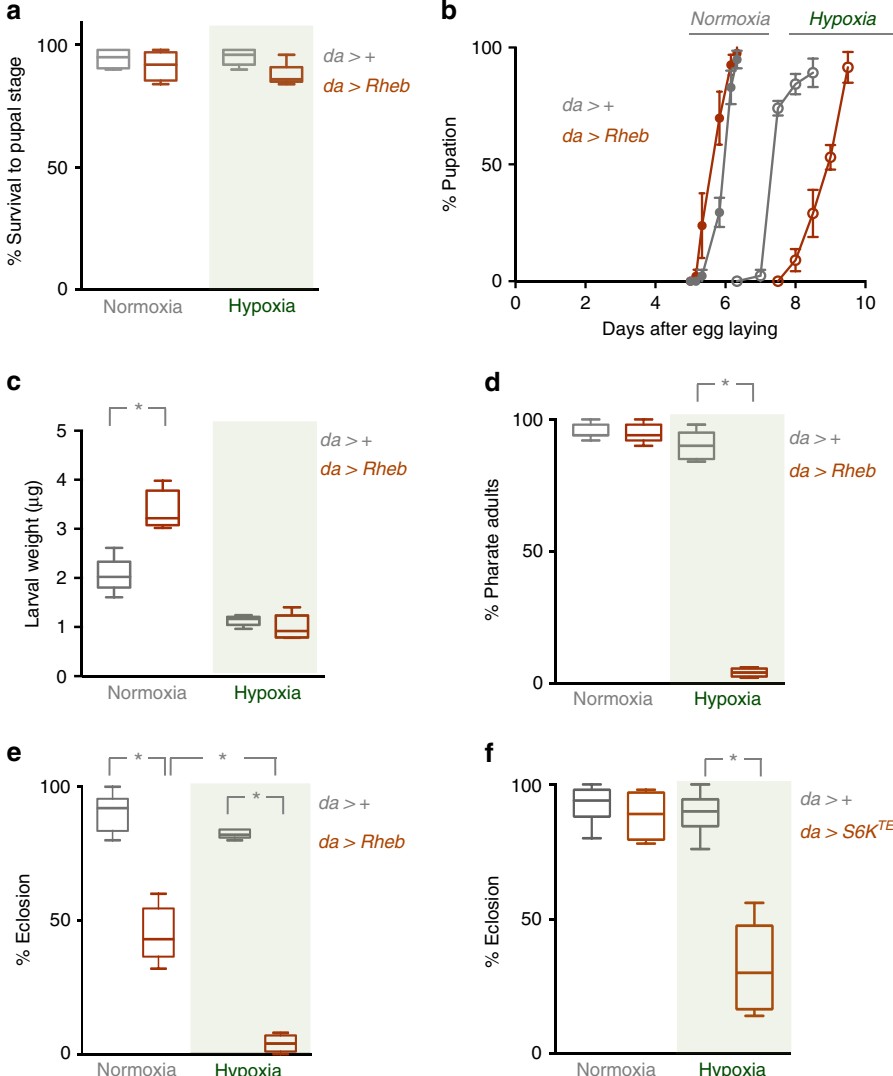

**Fig. 3** Suppression of TORC1 is required for adaptation to hypoxia. **a–d** Control (*da>+*) or Rheb overexpressing (*da > Rheb*) animals were maintained in either normoxia or hypoxia (5% oxygen) throughout the larval period. **a** Survival to the pupal stage was measured by calculating the percentage or larvae that developed to pupae for each experimental condition. **b** The rate of larval development was measured by calculating the percentage of animals that progressed to the pupal stage over time. $N =$ a minimum of four independent groups of animals (50/group). Maintaining TORC1 signalling in larvae during hypoxia led to a further delay in pupation. **c** Weights of wandering third instar larvae were measured for each experimental condition. Maintaining TORC1 signalling in hypoxia did not increase larval growth. Data are presented as box plots (25%, median and 75% values) with error bars indicating the min and max values. $N = 5$ groups of larvae per condition. **d**, **e** Control (*da>+*) or Rheb overexpressing (*da > Rheb*) animals were maintained in either normoxia or hypoxia (5% oxygen) throughout the larval period, before being returned to normoxia at the pupal stage. The percentage of animals that developed to **d** pharate adults, or **e** eclosed adults were then calculated. Maintaining TORC1 signalling in larvae during hypoxia led to a subsequent lethality during pupal development. Data are presented as box plots (25%, median and 75% values) with error bars indicating the min and max values. $N =$ five independent groups of animals (50 animals per group) per experimental condition. *$p < 0.05$, Students *t*-test. **f** Control (*da>+*) or S6K^TE overexpressing (*da > S6K^{TE}*) animals were maintained in either normoxia or hypoxia (5% oxygen) throughout the larval period, before being returned to normoxia at the pupal stage. The percentage of animals that developed to adults was then calculated. Data are presented as box plots (25%, median and 75% values) with error bars indicating the min and max values. $N =$ a minimum of four independent groups of animals (50 animals per group) per experimental condition. *$p < 0.05$, Students *t*-test

wandering third instar larval weight (Fig. 3c) and pupal volume (Supplementary Fig. 8a). These results are consistent with increased growth caused by modest elevation of TORC1 signalling. However, we found that when raised in hypoxia, the increase in size in $da > Rheb$ animals was abolished (Fig. 3c, Supplementary Fig. 8a). Given that the $da > Rheb$ pupae required an additional ~2 days of larval development to reach the same size as the $da>+$ animals, this indicates that the Rheb overexpressing animals actually had a reduced growth rate in hypoxia.

Finally, we examined how maintaining TORC1 activity during larval development in hypoxia affects subsequent survival to adulthood. For these experiments, we maintained animals in either normoxia or hypoxia throughout their larval period and then switched them to normoxia at the end of the larval period and monitored their subsequent development and viability. We first saw that animals carrying either the $da$-$Gal4$ ($da>+$) or UAS-Rheb ($+>Rheb$) transgenes alone, or animals expressing a UAS-GFP transgene ($da > GFP$) had no differences in viability after either normoxia or hypoxia exposure (Supplementary Fig. 8b,c). We found that both $da>+$ and $da > Rheb$ animals grown in normoxia as larvae showed normal development to the pharate adult stage. Similarly, $da>+$ animals grown in hypoxia as larvae also showed no significant change in development to pharate adults. In contrast, $da > Rheb$ animals that were maintained in hypoxia during their larval period showed a marked lethality at the pupal stage with few animals developing to pharate adults (Fig. 3d). When we further examined adult eclosion, we again saw that $da > Rheb$ animals that were maintained in larval hypoxia showed almost complete lethality, but in this case the $da > Rheb$ animals raised in normoxia also showed a reduction in eclosion, albeit to a much lesser extent than their hypoxia-raised counterparts. We repeated our Rheb overexpression experiments with a second independent $UAS$-$Rheb$ transgene and we observed similar, but slightly weaker effects, where $da > Rheb$ animals grown in hypoxia as larvae showed a significant decrease in survival to adult stage compared to $da>+$ animals (Supplementary Fig. 8d).

One key downstream effector of TORC1 is ribosomal protein S6 kinase (S6K), which is directly phosphorylated and activated by TORC1. Since we saw that hypoxia suppressed S6K phosphorylation, we tested whether reduced S6K activity was also required for hypoxia tolerance. To do this we examined the effects of ubiquitous expression of a constitutively active version of S6K (UAS-S6K$^{TE}$) using $da$-$Gal4$. We found that both control ($da>+$) and S6K$^{TE}$ expressing ($da > S6K^{TE}$) animals grown in normoxia as larvae showed normal development to the adult stage. Similarly, $da>+$ animals grown in hypoxia as larvae also showed no significant change in development to pharate adults. However, as with Rheb overexpression, the $da > S6K^{TE}$ larvae that were maintained in hypoxia showed a marked lethality at the pupal stage with few animals eclosing as adults (Fig. 3f). Taken together, these experiments indicate that suppression of both TORC1, and its effector S6K, are required for larvae to reset their development and growth rate in hypoxic environments, and for subsequent viable development to the adult stage.

The adaptation to hypoxia may reflect a cell-autonomous requirement for each cell to sense low oxygen and inhibit TORC1 to promote overall development and survival. Alternatively, hypoxia may modulate TORC1 in one particular tissue to control overall body growth and development. A precedent for this is the nutrient regulation of larval physiology and growth. For example, nutrient-dependent changes in TORC1 signalling in specific tissues such as the fat body or prothoracic gland can control whole animal growth and development through non-autonomous effects on endocrine signaling[9,10]. In this manner, one tissue

functions as a sensor of environmental stimuli to coordinate whole-body responses. To examine a potentially similar role in hypoxia sensing, we examined whether TORC1 suppression in a specific tissue was required for hypoxia tolerance in developing *Drosophila*. To do this we again took the approach of expressing a $UAS$-$Rheb$ transgene to maintain TORC1 signaling under hypoxia, but this time we restricted Rheb expression to specific larval tissues. We chose to examine effects on hypoxia tolerance by maintaining animals in either normoxia or hypoxia during their larval period and then measuring survival to eclosion. We tested Gal4 drivers that express in the fat body ($r4$-$Gal4$), neurons ($elav$-$Gal4$), the intestine ($MyoIA$-$Gal4$), the prothoracic gland ($P0206$-$Gal4$) and the muscle ($dmef2$-$Gal4$). We found the most dramatic effects were seen with fat-specific expression of Rheb: $r4 > Rheb$ animals grown in hypoxia during their larval stage showed a significant decrease in adult survival compared to both $r4>+$ control animals (Fig. 4a) and $r4 > GFP$ control animals (Supplementary Fig. 9a). In contrast to ubiquitous expression of Rheb, we found that fat body restricted expression of Rheb did not delay larval development in hypoxia—$r4 > Rheb$ animals developed slightly faster to the pupal stage in both normoxia and hypoxia compared to control ($r4>+$) animals (Supplementary Fig. 9b). Also, $r4 > Rheb$ animals showed no significant change in final pupal size compared to $r4>+$ animals (Supplementary Fig. 9c). To further explore the role for suppression of TORC1 signaling in the fat body on hypoxia survival, we examined the effects of RNAi-mediated knockdown of TSC2. As with Rheb overexpression, we found that $r4 > TSC2$ $IR$ animals showed a significant reduction in survival to the adult stage when maintained in hypoxia throughout their larval development (Fig. 4b).

In contrast to these fat body effects on hypoxia survival, when we expressed Rheb in either neurons, intestine or prothoracic gland we saw no effect on viability in animals exposed to hypoxia (Fig. 4c–e). Animals expressing Rheb in muscle ($dmef2 > Rheb$) did show reduced adult survival when grown in hypoxia as larvae; however, they also showed reduced survival in normoxia, making the effects on hypoxia tolerance difficult to interpret (Fig. 4f).

These results suggest that the larval fat body is an important hypoxia-sensing tissue that responds to low oxygen by suppressing TORC1 activity to ensure subsequent viable development. We therefore focused our attention on understanding how reduced TORC1 signaling in the fat body contributes to hypoxia tolerance.

## Reduced TORC1 increases lipid droplet size and lipid storage.

We next examined how reduction of TORC1 signaling in the larval fat body contributes to normal organismal development and survival in hypoxia. The role of the fat body as a coordinator of overall body physiology and development has been best studied in the context of altered dietary nutrients. In particular, when larvae are starved of nutrients the fat body mobilizes stored sugars and lipids in order to maintain circulating levels of these nutrients and support tissue homeostasis[9,10]. Upon starvation, fat body cells also rapidly engage autophagy to promote organismal survival[17]. We therefore examined whether these changes are associated with exposure to low oxygen. We first examined autophagy since this is a well-studied conserved process known to be induced by TORC1 inhibition[17]. We subjected early third instar larvae to hypoxia for 6 h and then stained fat bodies with Lyso-Tracker Red to visualize lysosomes and late stage autophago-somes as an indicator of autophagy. We also stained fat bodies from larvae maintained in normoxia and from larvae subjected to 6 h of nutrient starvation, a condition known to induce autophagy. We found that fat bodies from normoxic animals showed

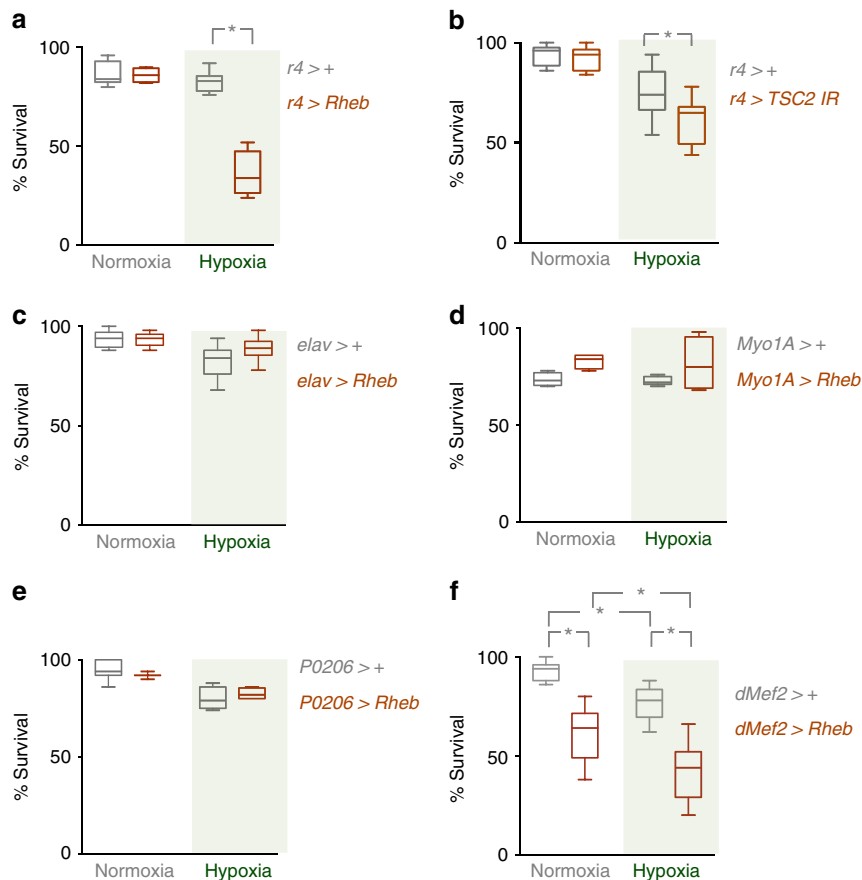

**Fig. 4** Suppression of TORC1 in the fat body is required for adaptation to hypoxia. **a** Control (r4>+) larvae or larvae overexpressing Rheb in the fat body (r4 > Rheb) were maintained in either normoxia or hypoxia (5% oxygen) throughout the larval period and then were returned to normoxia at the beginning of the pupal stage. The percentage of animals that survived to adults was then measured. Animals expressing Rheb in the fat body and exposed to hypoxia as larvae showed a significant decrease in adult survival. Data are presented as box plots (25%, median and 75% values) with error bars indicating the min and max values. $N =$ a minimum of four groups (50 animals per group) per experimental condition. $*p < 0.05$, Students $t$-test. **b** Control larvae (r4>+) or larvae expressing an inverted repeat RNAi transgene to TSC2 (r4 > TSC2 IR) were maintained in hypoxia throughout the larval period from hatching to pupation. Animals were then returned to normoxia and the percentage of animals surviving to the adult stage counted. Data are presented as box plots (25%, median and 75% values) with error bars indicating the min and max values. $N = 6$ groups of animals (50 animals per group) per experimental condition. $*p < 0.05$, Students $t$-test. **c**–**e** Rheb was overexpressed in larval neurons (**c**, elav > Rheb), the intestine (**d**, MyoIA > Rheb), the prothoracic gland (**e**, P0206 > Rheb) or muscle (**f**, dmef2 > Rheb). Animals were hatched in normoxia and then maintained throughout the larval period in either normoxic or hypoxic conditions, before being returned to normoxia at the pupal stage. The percentage of animals that survived to the adult stage was calculated for each experimental condition. Control animals carried the Gal4 transgene alone. Data are presented as box plots (25%, median and 75% values) with error bars indicating the min and max values. $N =$ a minimum of four independent groups of animals (50 animals per group) per experimental condition. $*p < 0.05$, Students $t$-test

no punctae staining with LysoTracker Red, while starved fat bodies showed a marked increase in LysoTracker Red punctae, consistent with induction of autophagy (Fig. 5). In contrast, we saw no LysoTracker Red punctae in fat bodies from larvae exposed to hypoxia for 6 h (Fig. 5). Even longer hypoxia exposure (24 h) also did not induce autophagy.

We then explored effects of hypoxia on lipid metabolism. In the fat body, triacylglycerols (TAGs) are stored within large lipid droplets. These lipid stores then can be mobilized under starvation conditions to supply a source of free fatty acid for beta-oxidation and other metabolic processes required for homeostasis[36]. We observed that when larvae were raised in hypoxia they showed a noticeable change in fat body morphology, which became less opaque in appearance as has been reported previously[37]. When we examined the fat bodies under light microscopy we saw an increase in cytoplasmic lipid droplet size (Fig. 6a). We examined this phenotype in more detail by using Nile Red to stain the neutral lipids that compose these cytoplasmic lipid droplets. When we transferred second instar

larvae grown in normoxia to hypoxia for 2 days we observed a significant increase in lipid droplet diameter compared to larvae maintained in normoxia for the same period (Fig. 6b, c). This effect on lipid droplets was opposite to that seen in larvae that were starved of all nutrients for 2 days (PBS only), which exhibited a marked decrease in lipid droplet size (Fig. 6b). Instead, the hypoxia phenotype was similar to animals that were transferred to a sugar-only diet for 2 days (Fig. 6b). These results indicate that the effects of hypoxia on lipid droplet size are opposite to those seen in nutrient deprivation and suggest that under hypoxia larvae may increase TAG levels through increased synthesis from dietary sugars. To measure TAG levels more quantitatively, we raised larvae from hatching in either normoxia or hypoxia and then measured whole-body TAG levels using a colorimetric assay. We found that hypoxic animals exhibited approximately a two-fold increase in total TAG levels when corrected for total larval weight (Fig. 6d). We additionally used a previously described sucrose solution buoyancy assay to estimate larval lipid content[38,39]. In this assay groups of isolated

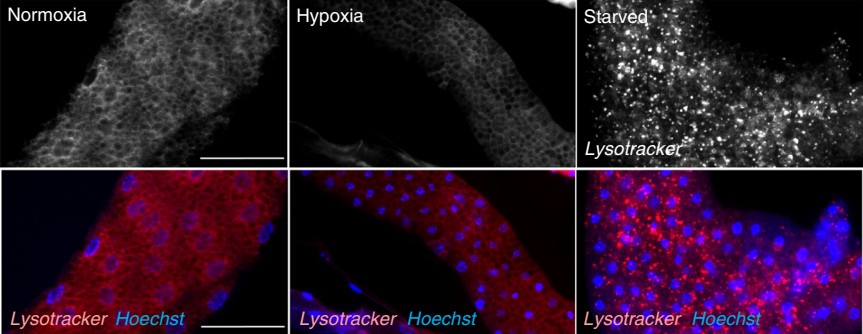

**Fig. 5** Hypoxia does not induce autophagy. Early third instar larvae were maintained in normoxia (left panels), transferred to hypoxia for 6 h (middle panels), or starved on PBS for 6 h (right panels). Fat bodies were then dissected, stained with LysoTracker and then imaged. Red = LysoTracker; blue = Hoechst DNA stain. Scale bar = 50 μm. Starvation, but not hypoxia, induced autophagy in the fat body. For each condition, fat bodies were dissected from at least fifteen independent larvae. The images are representative lysotracker staining from each condition. We observed essentially no LysoTracker Red stained punctae in either the fed or hypoxic fat bodies, but all the starved fat bodies showed pronounced staining similar to that seen in the representative image in this figure

wandering third instar larvae are mixed with increasing concentrations of a sucrose solution and the percentage of larvae floating at each concentration is measured. Using this approach, we found that hypoxic larvae were more buoyant than larvae grown in normoxia, consistent with an increase in lipids as a proportion of total body mass (Fig. 6e). Altogether, these results indicate that hypoxia induces a remodelling of lipid droplets and an increase in total lipid storage.

We next examined whether these changes in lipid metabolism occurred as a consequence of reduced TORC1 activity. To test this, we generated GFP-marked fat body *tsc1* mutant cell clones. As we previously described, loss of TSC1 completely reversed the hypoxia-mediated suppression of TORC1 signaling. Hence, we examined these *tsc1* mutant fat body cells to see if they still showed the hypoxia-mediated changes in lipid droplets. We induced clones during mitosis in the embryo and then when the animals hatched we transferred them to hypoxia for the rest of their larval development. When we dissected and examined the fat bodies from hypoxic third instar larvae using DIC microscopy, we observed the hypoxia increase in lipid droplet size in all non-GFP cells (Fig. 7). However, the *tsc1* mutant cells showed no increase in lipid droplet size. Instead they maintained the small lipid droplet morphology typical of normoxic animals at the same stage even though the animals had been grown in hypoxia for several days (Fig. 7, Supplementary Fig. 10a). These data indicate that suppression of TORC1 signalling is required for the hypoxia-mediated remodelling of lipid storage.

**Altered lipid metabolism is required for hypoxia tolerance**. We next examined whether the changes in lipid storage caused by the hypoxia-mediated suppression of fat body TORC1 signaling were important for development and survival. To do this we tested how fat-body specific knockdown of different genes important for lipid metabolism would affect the ability of animals to tolerate larval hypoxia and survive to the adult stage. We first examined genetic knockdown of Lsd2, a *Drosophila* perilipin homolog[40–42]. Lsd2 is a protein associated with the surface of lipid droplets that is necessary for normal lipid droplet formation. We used expression of an inverted repeat (IR) to Lsd2 (*UAS-lsd2 IR*) to specifically knockdown Lsd2 in the fat body using the *r4-Gal4* driver. When we did this and then transferred animals to hypoxia for 2 days, we found that the large lipid droplet phenotype seen in control (*r4>+*) animals was blocked when Lsd2 levels were reduced (*r4 > lsd2 IR*) (Fig. 8a, Supplementary Fig. 10b). We then explored how this inhibition of lipid droplet size affected tolerance to hypoxia. We maintained *r4>+* and *r4 > lsd2 IR* larvae in

hypoxia from larval hatching to pupation, and then switched them back to normoxia and monitored viability to adult stage. We found that the *r4 > lsd2 IR* showed a significant reduction in survival compared to *r4>+* control animals (Fig. 8b). To confirm this effect, we also examined a previously reported *lsd2* mutant allele (*lsd2KG00149*)[42]. These *lsd2* mutants are viable and show normal development when grown on normal laboratory food in normoxia. However, when we maintained these *lsd2* mutants in hypoxia throughout their larval period, they showed a marked reduction in survival to adult stage compared to control (*w1118*) animals (Fig. 8c). We next examined how altering lipid synthesis affects survival in hypoxia. To do this we tested the effects of fat body specific RNAi-mediated knockdown of two genes—acetyl-CoA carboxylase (ACC), an enzyme required for the first step of conversion of acetyl-CoA to fatty acids, and lipin, an enzyme with both phosphatase and transcriptional activity required for lipid synthesis. We found that, *r4 > ACC IR* and *r4 > lipin* IR animals showed little effect on viability when raised in normoxia (Supplementary Fig. 10c), but exhibited a significant reduction in viability to adult stage when they were maintained in hypoxia throughout their larval development (Figs. 8d,e). Finally, we examined the effects of fat body specific (*r4-Gal4*), overexpression of brummer (*bmm*), a triglyceride lipase. We found that *bmm* overexpression blocked the increase in lipid droplet size seen in fat body cells when larvae were exposed to hypoxia (Fig. 8a, Supplementary Fig. 10b). In addition, we saw that, when compared to control *r4>+* animals, *r4 > bmm* animals showed normal survival when raised in normoxic conditions (Supplementary Fig. 10c), but showed a significant reduction in viability to adult stage when they were maintained in hypoxia throughout their larval development (Fig. 8f). These results together show that the increase in lipid droplet size caused by reduced TORC1 is required for organismal adaptation to hypoxia.

## Discussion

In this paper, we explore how *Drosophila* are able to tolerate hypoxia. A central finding of our work is that when larvae are exposed to low oxygen, the fat body serves as a key hypoxia sensor that mediates changes in physiology to ensure viable organismal development. This hypoxia sensor role is mediated through inhibition of TORC1 signaling and reorganization of lipid storage. This function of the fat body as a hypoxia sensor is similar to the role of the fat body in coordinating whole-body physiology responses to changes in dietary nutrients[9,10,17,43,44]. As we find in hypoxia, these nutrient effects are also dependent on modulation of TORC1 activity and they can exert both

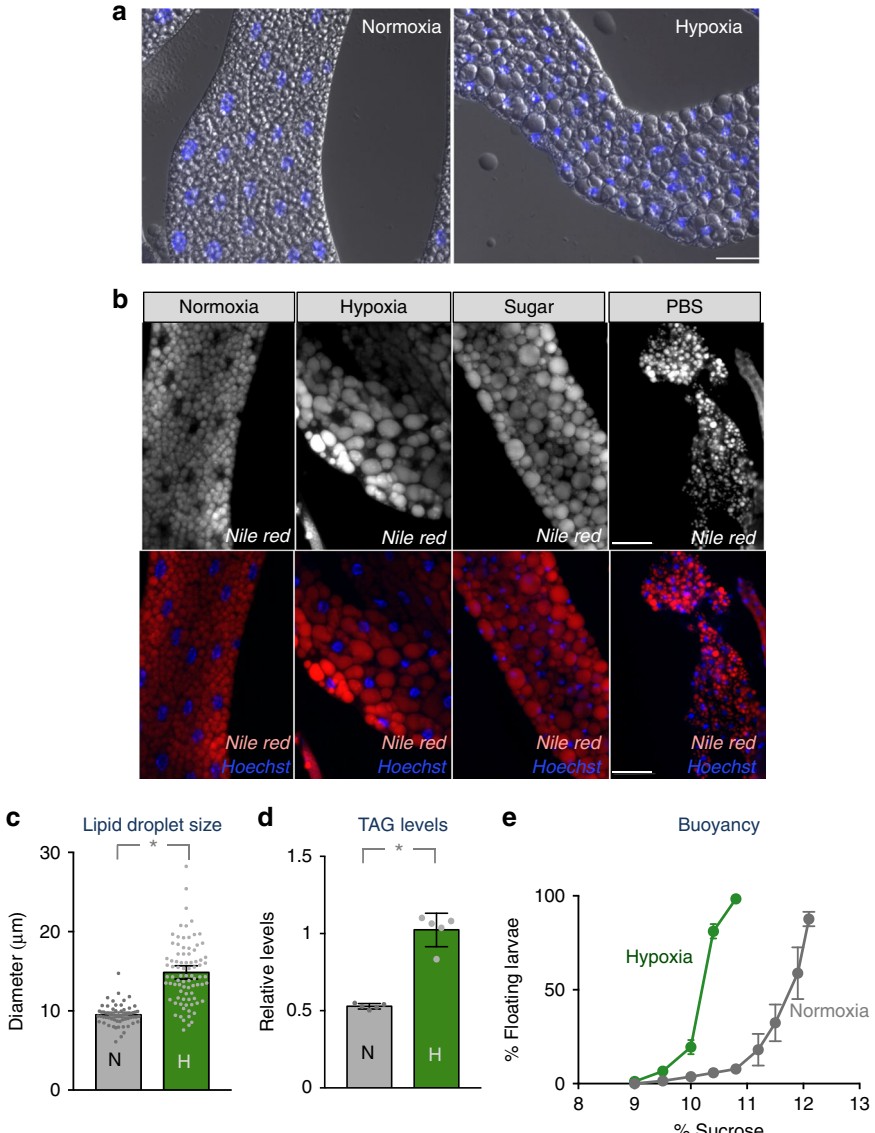

**Fig. 6** Hypoxia alters lipid levels and lipid storage. **a** Larvae were hatched in normoxia and then maintained throughout the larval period in either normoxic or hypoxic conditions. Fat bodies from third instar larvae were then imaged using DIC microscopy. Blue = Hoechst DNA stain. Hypoxia led to an increase in lipid droplet size in fat body cells. Scale bar = 50 μm **b** $w^{1118}$ larvae were grown in normoxia until 72 h after egg laying at which point they were transferred to one of four experimental conditions for 48 h: normoxia, hypoxia, sugar-only diet, complete starvation (PBS) diet. Fat bodies were then dissected, stained with Nile Red and then imaged. Red = Nile Red, blue = Hoechst DNA dye. **c**. Lipid droplet sizes from normoxic and hypoxic fat bodies represented in B were measured and presented as mean diameter ±SEM (*$p < 0.05$, Students $t$-test). **d** Larvae were hatched in normoxia and then maintained throughout the larval period in either normoxic or hypoxic conditions. Total TAG levels from third instar larvae from both experimental conditions were then measured. Data are presented as mean ±SEM (*$p < 0.05$, Students $t$-test). **e** Larvae were hatched in normoxia and then maintained throughout the larval period in either normoxic or hypoxic conditions. Larval lipid content was then estimated in wandering third instar larvae using an assay, which measures the percentage of larvae that float in increasing amounts of a sucrose solution. $N = 5$ independent groups of larvae (50/group)

metabolic and endocrine effects to control growth and development. These studies, and our findings in hypoxia, emphasize how the fat body functions as a sentinel tissue to detect changes in environmental conditions and to buffer the internal milieu from these changes. Moreover, while most work on hypoxia has focused on studying cells in culture[27,45], our findings emphasize the importance of non-cell autonomous mechanisms in controlling how animals adapt to low oxygen.

Inhibition of TORC1 in larvae exposed to hypoxia occurred rapidly and, interestingly, only in response to a specific range of low oxygen (~2–6%). At 2% oxygen and lower, the response to hypoxia is very different compared to exposure to 5% oxygen that was used in this study—larvae crawl away from the food and

eventually undergo complete movement arrest, which can be reversed within minutes of return to normoxia. Larvae can only tolerate this level of low oxygen (<2%) for a few hours before dying. Since this low oxygen hypoxic response is different to the behaviour of larvae at 5% oxygen (which maintain their feeding and growth) it may also rely on qualitatively different changes in hypoxia sensing and signaling that do not involve suppression of TORC1.

The hypoxia-mediated inhibition of TORC1 that we found required TSC1/2, but was independent of one main mechanism defined in mammalian cell culture experiments: induction of REDD1 by the well-studied HIF-1 alpha transcription factor. Although the *Drosophila* homolog of REDD1, Scylla, was

**Fig. 7** TORC1 suppression is required for hypoxia-induced modulation of lipid storage. The MARCM system was used to generate GFP-marked *tsc1*[W240X] mutant cell clones in the fat body. Hatched larvae were then either maintained in normoxia or transferred to hypoxia. At the third instar stage, larval fat bodies were fixed, dissected and mounted on coverslips. Fat bodies were then imaged using DIC microscopy to visualize lipid droplets. Blue = Hoechst DNA dye. Scale bar = 50μm

previously shown to be sufficient to inhibit TORC1[32], we found that it was not necessary. Indeed, analysis of the REDD1 mutant mouse also showed that in certain tissues, hypoxia-mediated repression of TORC1 was also REDD1-independent[34]. Our results also suggest that other signaling molecules, such as AMPK, Ptp61F and nitric oxide, may not be involved in hypoxia-mediated suppression of TORC1 signaling. A previous report in cell culture showed that upon different stresses including hypoxia, TSC2 could translocate to the lysosome and inhibit Rheb activation of TORC1[46]. Therefore, upon hypoxia exposure in larvae, the TSC1/2 complex may rapidly relocalize to inhibit TORC1 function. The mechanism that could drive this (or any other potential mechanism of TORC1 inhibition) must be triggered rapidly in response to hypoxia in larvae. Given the importance of oxygen as an electron acceptor in the electron transport chain in the mitochondria, it is plausible that the rapid sensing of low oxygen in larvae occurs as a result of altered mitochondrial activity. Two potential hypoxia effectors in this scenario are induction of reactive oxygen species or alterations in the levels of mitochondrial metabolites. For example, both 2-hydroxyglutarate and alpha ketoglutarate—metabolites in the TCA cycle—can alter TORC1 activity[47,48] and both can be induced by hypoxia in mammalian cell culture experiments[49]. How these changes could then subsequently lead to an increase in TSC1/2 function and/or localization to inhibit TORC1 remains to be determined. Also, although our genetic data indicate that TSC1/2 is required for hypoxia suppression of TORC1, we cannot rule out the possibility that other non-dominant pathways may also act to influence TORC1 signaling in low oxygen.

We found that the protective effect of lowering adipose TORC1 signaling in hypoxia involved the reorganization of lipid metabolism. Hypoxic larvae increase their fat body lipid droplet size and their total proportion of whole-body TAGs. This response is very different from starvation, a stress that also inhibits TORC1, where lipid droplet size is markedly reduced. This has been reported to be due to upregulation of lipases such as the ATGL lipase, Brummer[50–52]. This mechanism provides a way to generate a source of free fatty acids for beta-oxidation to maintain homeostasis and to allow survival under scarce nutrient conditions. In our experiments, the hypoxia-induced lipid droplet phenotypes were similar to those seen when larvae were switched to a sugar-only diet. One interpretation of this result is that under hypoxia larvae mobilize glucose toward new TAG lipid synthesis and storage. Hence, the large lipid droplets and increased TAG levels seen in hypoxia may be primarily due to new TAG synthesis rather than a suppression of lipolysis. Indeed, we find that at least two enzymes involved in lipid synthesis (ACC and Lipin) are required for the hypoxia-mediated increase in lipid droplet size. Whether these metabolic effects rely on conversion of dietary glucose to lipids or involve mobilization of fat body glycogen stores to provide glucose for de novo lipid synthesis remains to be determined. Future experiments tracking glucose metabolic flux in

hypoxic larvae would be useful to resolve these points. Our data also suggest that the effect of TORC1 inhibition on lipid storage differs between stress stimuli that suppress TORC1 (hypoxia vs starvation). Indeed, we also saw that decreased TORC1 activity in hypoxia did not trigger autophagy in the fat body in the same way that it does upon nutrient deprivation.

One result we found interesting was that the suppression of fat body TORC1 and altered larval lipid storage was not necessary for viable larval development under hypoxia, but was required for subsequent development in the pupal stage to produce viable adults. During the pupal stage, tissues undergo metamorphosis to establish the adult body. Since this is also a non-feeding stage of the life cycle, the energy required to fuel these extensive tissue rearrangements in pupae must therefore come from stored nutrients. It has been calculated that the lipid stores provide 90% of this energy[53]. Our findings suggest that pupae may be more dependent on these lipid stores after a period of prior larval hypoxia. Hence failure to maintain these stores, either by preventing TORC1 inhibition (Rheb over-expression) or genetic disruption of lipid droplet formation (Lsd2 knockdown), led to reduced viability in hypoxia, while having no effect on normal development in normoxia. It is also possible that the requirement for altered lipid stores may reflect a role for lipid droplets beyond simply providing a usable energy source[54]. A pertinent example is a report describing how increases in glial lipid droplets in larvae were important for maintaining neuroblast cell proliferation in larvae exposed to hypoxia or oxidative stress[55]. In this case, the lipid droplets were required to play an antioxidant role to buffer neurons from ROS-induced damage. Mammalian cancer cells in culture have also been shown to accumulate lipid droplets in low oxygen, an effect that is important to promote their survival and tumorigenic phenotypes in mouse models[56]. Cancer cells with high levels of TORC1 activity are also dependent on exogenous fatty acids for their survival in hypoxic conditions[57,58]. Finally, a recent study in *Drosophila* also showed that the type of dietary lipid source—plant versus yeast —was important in determining whether flies could tolerate cold stress, where plant-based unsaturated fatty acids were able to maintain membrane fluidity and locomotor activity at low temperatures[59]. The lipid changes we observed in hypoxia may, therefore, be important for ensuring cell and tissue viability in pupal stages independent of any role in energy production.

One potential prediction of our work is that lowering TORC1 may provide a protective response against the damaging effect of low oxygen exposure. This is difficult to test in larvae since low TORC1 decreases overall growth and development in normal conditions. However, given the importance of both the fat body and TORC1 as regulators of adult *Drosophila* physiology, particularly in the context of aging and stress responses, the larval hypoxia responses we describe here provide a general mechanism of low oxygen tolerance throughout *Drosophila* life.

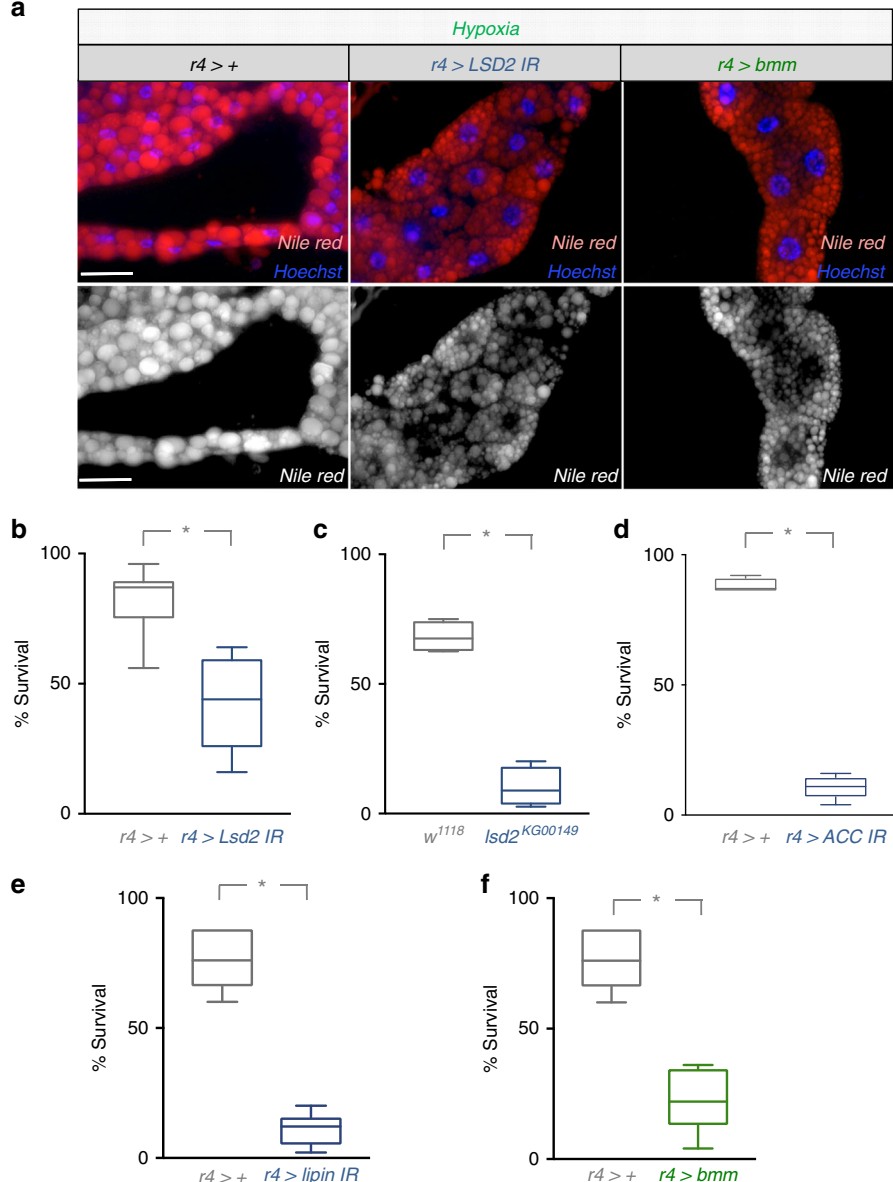

**Fig. 8** Reorganization of lipid droplets is required for adaptation to hypoxia. **a** Control larvae (r4>+) larvae expressing an RNAi transgene to Lsd2 (r4 > Lsd2 IR), or larvae expressing *UAS-brummer* (r4 > bmm) were exposed to hypoxia for 48 h of hypoxia and fat bodies were stained with Nile Red. Scale bar = 50μm. **b** Control larvae (r4>+) or larvae expressing an inverted repeat RNAi transgene to Lsd2 (r4 > Lsd2 IR) were maintained in hypoxia throughout the larval period from hatching to pupation. Animals were then returned to normoxia and the percentage of animals surviving to the adult stage was counted. N = 5 groups of animals (50 animals per group) per experimental condition. *p < 0.05, Students t-test. **c** Control (w[1118]) or lsd2 mutant (lsd2[KG00149]) larvae were maintained in hypoxia throughout the larval period from hatching to pupation. Animals were then returned to normoxia and the percentage of animals surviving to the adult stage counted. N = 4 independent groups of animals (50 animals per group) per experimental condition. *p < 0.05, Students t-test. **d**, **e** Control larvae (r4>+) or larvae expressing an inverted repeat RNAi transgene to either **d** ACC (r4 > ACC IR) or **e** lipin (r4 > lipin IR) were maintained in hypoxia throughout the larval period from hatching to pupation. Animals were then returned to normoxia and the percentage of animals surviving to the adult stage was counted. N = 4 independent groups of animals (50 animals per group) per experimental condition. *p < 0.05, Students t-test. **f** Control larvae (r4>+) or larvae expressing *UAS-brummer* (r4 > bmm) were maintained in hypoxia throughout the larval period from hatching to pupation. Animals were then returned to normoxia and the percentage of animals surviving to the adult stage counted. N = 6 independent groups of animals (50 animals per group) per experimental condition. *p < 0.05, Students t-test. For panels **b**–**f**, the data are presented as box plots (25%, median and 75% values) with error bars indicating the min and max values

## Methods

***Drosophila* stocks**. Flies were raised on standard medium containing 150 g agar, 1600 g cornmeal, 770 g Torula yeast, 675 g sucrose, 2340 g D-glucose, 240 ml acid mixture (propionic acid/phosphoric acid) per 34 L water and maintained at 25 °C, unless otherwise indicated. The following fly stocks were used:

w[1118], tsc1[Q87X]/TM6B[60], sima[07607]/TM3,Ser,GFP[21], scylla/TM3,Ser,GFP[32], charybdis[180]/TM3,Ser,GFP[32], lsd2[KG00149] (Bloomington Stock Centre), UAS-ptp61F-RNAi (Bloomington Stock Centre), UAS-NOS-RNAi (VDRC), UAS-sima-

RNAi (VDRC), UAS-tsc2-RNAi (VDRC), UAS-AMPK RNAi (VDRC), UAS-Rheb (Bloomington Stock Centre), UAS-S6K-TE (Bloomington Stock Centre), UAS-Lsd2 RNAi (Bloomington Stock Centre), da-Gal4, r4-Gal4, P0206-Gal4, Elav-Gal4, MyoIA-Gal4, dmef2-GAL4.

For all GAL4/UAS experiments, homozygous GAL4 lines were crossed to the relevant UAS line(s) and the larval or adult progeny were analyzed. Control animals were obtained by crossing the relevant homozygous GAL4 line to flies of the same genetic background as the particular experimental UAS transgene line.

**Hypoxia exposure**. For all hypoxia experiments (except for those shown in Fig. 2b and Supplementary Fig. 2) *Drosophila* were exposed to 5% oxygen. This was achieved by placing vials containing *Drosophila* into an airtight glass chamber into which a mix of 5% oxygen/95% nitrogen continually flowed. Flow rate was controlled using an Aalborg model P gas flow meter. Alternatively, for some experiments *Drosophila* vials were placed into a Coy Laboratory Products in vitro $O_2$ chamber that was maintained at fixed oxygen levels of 1–20% (depending on the nature of the experiment: See Fig. 2b and Supplementary Fig. 2) by injection of nitrogen gas.

**Preparation of protein extracts**. *Drosophila* larvae were lysed with a buffer containing 20 mM Tris-HCl (pH 8.0), 137 mM NaCl, 1 mM EDTA, 25% glycerol, 1% NP-40 and with following inhibitors 50 mM NaF, 1 mM PMSF, 1 mM DTT, 5 mM sodium ortho vanadate ($Na_3VO_4$) and Protease Inhibitor cocktail (Roche Cat. No. 04693124001) and Phosphatase inhibitor (Roche Cat. No. 04906845001), according to the manufacturer instructions.

**Western blots, immunostaining and antibodies**. Protein concentrations were measured using the Bio-Rad Dc Protein Assay kit II (5000112). Protein lysates (20–40 μg) were resolved by SDS-PAGE and electro transferred to a nitrocellulose membrane, subjected to western blot analysis with specific antibodies, and visualized by chemiluminescence (enhanced ECL solution (Perkin Elmer)). For the anti-sima immunostaining experiments, larvae were inverted and fixed in 4% paraformaldehyde/PBS at room temperature for 45 min. Samples were blocked in 1% bovine serum albumin/ 0.1% Triton/PBS for 2 h and then incubated in primary sima antibody (1:200) overnight at 4°C. Samples were then washed 3x with 0.1% Triton X/PBS and incubated with Alexa 568 (Molecular probes) secondary antibody (1:4000 in 1% bovine serum albumin/ 0.1% Triton/PBS) for 2 h at room temperature. Samples were then mounted on slides using VectaShield mounting medium. Primary antibodies used in this study were: anti-phospho-S6K-Thr398 (1:1000, Cell Signalling Technology #9209), anti-eIF2alpha (1:1000, AbCam #26197), anti-S6K (1:2000, a gift from Aurelio Teleman), anti-actin (1:1000, Santa Cruz Biotechnology, # sc-8432), anti-Sima (1:200, a gift from Utpal Banerjee). Secondary antibodies were purchased from SantaCruz Biotechnology (sc-2030, 2005, 2020). For experiments looking at TORC1 activity, either total eIF2alpha, actin or total S6K levels were used as loading controls because the levels of these proteins were unaffected by hypoxia.

**Measurement of *Drosophila* development, growth and survival**. For measuring development timing to pupal stage, newly hatched larvae were collected at 24 h AEL and placed in food vials (50 larvae per vial). The number of pupae was counted each day. For each experimental condition, a minimum of five replicates was used to calculate the mean percentage of pupae per time point. To measure larval growth, newly hatched larvae were collected at 24 h AEL and placed in food vials (50 larvae per vial) and then maintained in either normoxia or hypoxia. Larvae were then imaged on each day of development using a Zeiss Discovery.V8 Stereomicroscope with Axiovision imaging software. For Fig 1A, images of larvae were taken on different days. The larval images were then cropped and arranged as shown in the figure. To measure larval weight, third instar larvae were washed in PBS, dried thoroughly on paper and then weighed in groups of ten using a microbalance.

Pupal Volume: Pupae were imaged using a Zeiss Discovery.V8 Stereomicroscope with Axiovision imaging software. Pupal length and width were measured and pupal volume was calculated using the formula, volume $= 4/3\pi(L/2)(l/2)^2$

**Larval and pupal imaging**. Larval and pupal images were obtained using a Zeiss Stereo Discovery V8 microscope using Axiovision software. Microscopy and image capture was performed at room temperature and captured images were processed using Photoshop CS5 (Adobe). To prepare final figures (Fig. 1, Supplementary Fig. 4), captured images of individual larvae and pupae were cropped and reoriented in Photoshop.

**LysoTracker and Nile red staining**. Fat bodies were dissected from larvae and incubated in either Nile Red (1:50,000 dilution of a 10% stock in DMSO, ThermoFisher Scientific, N1142) or LysoTracker (1:1000, Thermofisher Scientific, L7528) for 10 mins on glass slides. Fat bodies were then immediately imaged using a Zeiss Observer Z1 microscope using Axiovision software. For the lipid droplet quantifications, we followed a method previously described[61] in which the diameters of three largest lipid droplets in each of at least 30 cells per condition were measured.

**Lipid measurements**. Groups of ten larvae were washed in PBS, dried on filter paper and then weighed. Total TAG levels were determined using a colorometric assay as described in detail in ref. [62]. In brief, groups of ten larvae were lysed and lysates were heated at 70 °C for 10 min. Then they were incubated first with triglyceride reagent (Sigma; T2449) and then mixed with free glycerol reagent (Sigma; F6428). Colorimetric measurements were then made using absorbance at 540 nm. The calculated TAG levels were then corrected for larval weight to give a measure

of total TAG levels per microgram of larval weight. The buoyancy assay was carried out as described in detail in ref. [39]. In brief, groups of 50 wandering larvae for each condition were washed and transferred to an 8% sucrose/PBS solution in 50 ml conical tubes. Twenty percent sucrose solution was then added to the tubes in 1 ml increments to slowly increase the overall sucrose concentration. After each addition, the tubes were gently swirled and the numbers of floating larvae were counted.

**Quantitative PCR**. Total RNA was extracted using TRIzol according to manufacturer's instructions (Invitrogen; 15596–018). RNA samples were then subjected to DNase treatment according to manufacturer's instructions (Ambion; 2238 G) and reverse transcribed using Superscript II (Invitrogen; 100004925). The generated cDNA was used as a template to perform qRT–PCRs (ABI 7500 real time PCR system using SyBr Green PCR mix) using specific primer pairs. PCR data were normalized to either actin or Glyceraldehyde-3-phosphate dehydrogenase (GAPDH) levels. Each experiment was independently repeated a minimum of three times. The following primers were used:

*Sima*: forward: ACAGCATCATCAGCAGCAAC; reverse: GCTGCTGGAAAGTCCTGAAC

*Fatiga:* forward; GGAAACGGAAACAGGTCAAA; reverse; TGGTTCATGTCGCTGATGAT

*Alpha-tubulin*: forward; CCTTCGTCCACTGGTACGTT; reverse, GGCGTGACGCTTAGTACTCC

**Statistical analyses**. Data were analyzed by Students *t*-test, or two-way ANOVA followed by post-hoc students *t*-test where appropriate. All statistical analysis and data plots were performed using Prism software. In all figures, statistically significant differences are presented as: $*p < 0.05$.

**Reporting summary**. Further information on experimental design is available in the Nature Research Reporting Summary linked to this article.

## Data Availability

The source data underlying all figures are provided as a source data file. Any other information that support the findings of this study are available from the corresponding author upon reasonable request.

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

## Acknowledgements

We thank Paula Bellosta, Aurelio Teleman, Hugo Stocker, Ernst Hafen, Bruce Edgar, Utpal Banerjee and Iswar Hariharan for the gift of reagents and fly stocks. Stocks obtained from the VDRC, the NIG-Fly Stock Centre, Kyoto, Japan and the Bloomington *Drosophila* Stock Center (NIH P40OD018537) were used in this study. This work was supported by a CIHR operating grant and a NSERC Discovery grant to S.S.G. E.C.B. was supported by Alberta Innovates Health Solutions Graduate Studentship.

## Author Contributions

Project design and conceptualization: B.L., S.S.G.; Data collection and analysis: B.L., E.C.B., S.S.G.; Original manuscript draft: S.S.G.; Manuscript reviewing and editing: B.L., E.C.B., S.S.G. Supervision and funding acquisition: S.S.G.

## Additional information

**Competing Interests:** The authors declare no competing interests.

