## [Peer Review File · Nature Communications]

Editorial Note: Parts of this review have been redacted

Reviewers' comments:

Reviewer #1 (Remarks to the Author):

Key results: Lee et al., have identified that *Drosophila* larvae suppress TOR signaling as a mechanism to adapt to hypoxia. They find that hypoxia suppresses TOR signaling by TSC1/2 mediated inhibition of Rheb. Specifically, they observe that ectopic expression of Rheb in fat tissue interferes with the adaptation to hypoxia. Given the prominent role of fat cells in sensing and mediating systemic effects of hypoxia, they examine in further detail how fat cells modulate the response to reduced oxygen. Then they find is that hypoxia induces altered lipid metabolism. Finally they show that genetic manipulation of a *Drosophila* perilipin homolog is unable to adapt to hypoxia.

Originality and significance: While both the role of TOR signaling and lipid droplet accumulation have been implicated in hypoxia response, the authors here have identified a developmental window in a highly genetically tractable system in which the biology of hypoxia response can be probed further. These findings thus have the potential to enable further mechanistic discovery, as this study establishes a model with defined phenotypes. This study will thus be of interest to those studying the molecular physiology of hypoxia sensing and adaptation.

Data & methodology: The approach and methods adopted by the authors is largely of good quality. I have identified certain areas of improvements, some of which will be critical to provide further support to their model. In addition, I feel that the impact and value of this study to the scientific community will be much enhanced if some depth and richness can be added to this work (see comments in section below).

Conclusions: The manuscript is well written, clear and the authors interpret the results largely cautiously, except in some instances outlined below. While majority of the manuscript is clear again I have identified a few paras in results section which would benefit from editing.

Clarity and context: The abstract is clear and accessible and the authors have provided appropriate context in their discussion.

Suggested improvements:

1. The authors largely rely on the overexpression of UAS-Rheb as a method to manipulate TOR signaling. To increase the rigor of their work the following experimental additions will be crucial and I would encourage the authors to perform them:

a. Control: using *r4-Gal4* + is not a satisfactory control in physiology experiments (Fig 3 and 4). The authors should consider expressing a transgene such as GFP, RFP, LacZ etc., as a 'stuffer' in experiments where they compare to UAS-Rheb.

b. Knock-down based experiments: I wonder why the authors have not performed experiments with UAS-TSC1/TSC2 RNAi to complement the UAS-Rheb overexpression experiments. The lack of complementarity in experimental design reduces the rigor. I think it will be crucial for the authors to perform experiments with UAS-TSC1/TSC2 RNAi for critical datasets in Figure 3 and 4 (note: authors have mislabeled Figure 4 and called it Figure 3).

c. The model presented here will be significantly strengthened if the authors can test whether UAS-Rheb RNAi larvae have better outcomes in the hypoxia experiments performed in Figure 1. It will be exciting to test whether TOR inhibition either genetically or pharmacologically (Rapamycin) makes these larvae hypoxia resistant.

2. The model for the role of hypoxia in lipid remobilization will benefit from further rigorous testing as the authors only use *Lsd1-IR* as the only manipulation for this.

a. Figure 8: Also authors do not use appropriate controls for this, using *r4* + is not ideal as just expression of dsRNA in animals has physiological effects. Authors should use RNAi of a heterologous gene such as GFP-RNAi, LacZ or mCherry-RNAi as control.

b. While the authors have discussed Brummer lipase, they have not performed the crucial experiment of using UAS-*bmm* (over-expression) or UAS-*bmm*-RNAi. The prediction of their model would be that inhibition of this lipase should provide hypoxia resistance whereas over-expression of *bmm* should phenocopy *Lds2-IR*.

c. While the authors show that hypoxia phenocopies the effect of high sugar diet on Lipid droplets,

their claims and richness of the model will be significantly enhanced if they can test whether larvae which have been exposed to a high sugar diet or larvae of naturally occurring fat mutants (such as *adipose60*) survive hypoxia better.

On the whole the experiments presented seem rather unidirectional (either over-expression or knock down) and I would like to see the authors make improvements along these lines. For instance, experiments done in Figure 7 are more convincing and exhibit a depth lacking in Figure 3,4 and 8 from a genetic stand-point.

3. The data presented in Supplementary Figure 1, based on which the authors claim that hypoxia does not affect feeding is highly unsatisfactory and very subjective. The authors must present more convincing evidence to stake the claim that hypoxia does not affect food intake, else they need to leave this open-ended and inconclusive.

4. The sentences in results section whether authors use UAS-Rheb to ectopically activate TOR are written in quite a confusing manner. I would encourage the authors to edit this section to make it more readable, clear and logical. I had to read it multiple times to follow the argument made by the authors. Page 6 (lines: 14 onwards).

Reviewer #2 (Remarks to the Author):

The larval fat body is suggested to be a hypoxia-sensing tissue. Reduction of mTORC1 activity within it via a TS- mediated mechanism is required to reset the organism's global physiological response, thereby lowering its growth and metabolic rate. Evidence is presented that this involves re-modelling of lipid droplets and lipid storage.

Generally, the text is clear, figures are well presented and data convincing. The data would, in some cases however, be strengthened by combining data from repeat experiments, so that statistical significance can be established, eg in Figure 2 have the westerns been repeated and quantified? In Figures 5, 7, 8a, it is not clear how many larvae were studied and if possible quantification of differences should be added.

Some of the authors claims may be overstated, for example in Figure 2 – the authors claim that the control of mTORC1 activity by hypoxia is via TSC/Rheb. However, they do this by showing loss of TSC activity or overexpression of Rheb suppresses the phenotype. This does not exclude the possibility of a parallel non-dominant pathway being involved. They need to undertake molecular/biochemical studies to show this link exists.

The mechanistic insights provided by these findings using an in vivo model are significant and likely to be of interest to scientists from a broad range of fields. The analysis is strengthened by the genetic findings being reinforced by biochemical analysis. The authors also suggest that their results have significance to cancer growth in hypoxic conditions.

Minor points and typographical errors

Supp Figure 2b seems to show similar data to Figure 2b – could this data be combined?

Page 18 (13 lines from the bottom): suggest changing 'hypoxia larvae' to 'hypoxic larvae'

Page 19: Figure 4 is mislabelled as Figure 3

Reviewer #3 (Remarks to the Author):

Lee et al have analyzed the involvement of the TOR pathway in adaptation to hypoxia of *Drosophila* larvae. They found that TORC1 is inhibited in hypoxia and that this inhibition is essential for adaptation. TORC1 hypoxic inhibition depends on TSC1/2 but surprisingly, does not depend on

HIF/Sima, REDD1/Scylla/Charybdis or AMPK. Then, they examined in which organ(s) is TORC1 inhibition necessary for hypoxic adaptation, and found that it is required exclusively in the fat body. Then they found that TORC1 inhibition in the fat body does not trigger autophagy, but instead leads to lipid droplet enlargement and increased lipid storage. Finally, they showed that the above alterations in lipid metabolism at the fat body are sufficient for adaptation of *Drosophila* larvae to hypoxia. The manuscript reads very well, the experiments are in general technically sound and the conclusions have important implications for organismal adaptation to hypoxia. The results showing that, following TORC1 inhibition in the fat body, lipid metabolism reorganizes and mediates adaptation to hypoxia are convincing. The first part of the work showing that TORC1 inhibition is independent of Sima, Scylla/Charybdis and AMPK have some potential problems, and the authors need to perform additional experiments to further analyze TORC1 inhibition in hypoxia.

1) Scylla and Charybdis have been proposed to be partially redundant, so the anti S6K western blots of hypoxic larvae extracts should be repeated in a Scylla-Charybdis double mutant. Then we will know to what extent is TOR inhibition independent of these two genes.

2) Sima homozygous mutant larvae have been reported to die in hypoxia (although they are fully viable in normoxia). This reviewer suspects that there might be a problem with the sima mutant line. Please double-check this line and demonstrate that it is truly sima loss-of-function (the authors might want to utilize a Sima dependent transcriptional reporter and/or perform an anti-Sima western blot). Even if TORC1 inhibition depends on Sima, this manuscript will still provide highly valuable data, so I would be happy to recommend its publication.

Minor point: Please note that Figure 4 is mislabeled (Fig.3 instead of Fig 4).

Reviewer #1 (Remarks to the Author):

We thank the reviewer for their time and their thoughts about how to improve our paper. We've outlined our responses to their comments below:

Suggested improvements:

1. The authors largely rely on the overexpression of UAS-Rheb as a method to manipulate TOR signaling. To increase the rigor of their work the following experimental additions will be crucial and I would encourage the authors to perform them:

a. Control: using r4-Gal4> + is not a satisfactory control in physiology experiments (Fig 3 and 4). The authors should consider expressing a transgene such as GFP, RFP, LacZ etc., as a 'stuffer' in experiments where they compare to UAS-Rheb.

As suggested by the reviewer we have carried out experiments where we express UAS-GFP either ubiquitously (daGAL4) or in the fat (r4GAL4). In both cases expression of GFP as a 'stuffer' had no effect on hypoxia tolerance. These data are now presented in Supplemental Figs 8c and 9a.

b. Knock-down based experiments: I wonder why the authors have not performed experiments with UAS-TSC1/TSC2 RNAi to complement the UAS-Rheb overexpression experiments. The lack of complementarity in experimental design reduces the rigor. I think it will be crucial for the authors to perform experiments with UAS-TSC1/TSC2 RNAi for critical datasets in Figure 3 and 4 (note: authors have mislabeled Figure 4 and called it Figure 3).

We did try several different RNAi lines to tsc1 and tsc2 however we found that these mimicked the tsc1/tsc2 null mutant phenotypes and lead to larval growth arrest in normal conditions. This precluded our ability to test any effects on survival in hypoxia. However, we have added two other pieces of data:

- a) We did find one UAS-TSC2 RNA line that didn't have a growth arrest phenotype in normoxia when expressed in the fat body using r4GAL4. We tested this line for hypoxia survival and found that, like Rheb overexpression, knockdown of TSC2 lead to reduced hypoxia survival. This data is presented in Fig 4b
- b) We also tested S6K, a downstream component in the TORC1 pathway. When we expressed a constitutive activated version of S6K, we found that, like Rheb overexpression, activation of S6K lead to reduced hypoxia survival. This result is presented in Figure 3F.

These two new pieces data, together with our previous data with two independent UAS-Rheb expression lines (Figs 3, 4, Suppl Fig 8d), make a strong case that TSC-Rheb-TORC1-S6K signaling plays an important role in controlling hypoxia tolerance in *Drosophila*.

c. The model presented here will be significantly strengthened if the authors can test whether UAS-Rheb RNAi larvae have better outcomes in the hypoxia experiments performed in Figure 1. It will be exciting to test whether TOR inhibition either genetically or pharmacologically (Rapamycin) makes these larvae hypoxia resistant.

We agree with the reviewer that it would be exciting to test if decreasing TOR promotes hypoxia resistance. However, there are two confounding issues with the experiment in larvae. The first is that the larvae already show a high degree of hypoxia tolerance. We find that in our experiments, ~80-90% of control animals survive larval hypoxia. Hence, it is difficult to test for any further increase in survival that might be conferred by lowering TOR. The second challenge is that lowering TOR activity blocks growth in larvae maintained under normal (normoxic conditions). We found that even in normoxic, control conditions expression of Rheb RNAi (two independent UAS lines) or expression of TSC1 and TSC2, lead to growth arrest in almost all larvae, and the few animals that did pupate showed severely delayed development and did not eclose. Similarly, we found that rapamycin feeding also lead to marked decreases in growth and survival in normoxic conditions. Hence, exploring the effects of low TOR on hypoxia survival in larvae is not really possible.

Nevertheless, the idea of reduced TOR as a protective effect against hypoxia is an interesting idea. And, as we have now discussed in the paper, one potential future line of enquiry will be to test this not in larvae, but adults. Here, confounding effects on growth and development are not an issue, and there has been a lot of work on using both rapamycin and genetic suppression of TOR to look at responses to stress and lifespan in adults. These experiments are beyond the scope of the current paper, but we could predict that lowering TOR in adults could protect against the effects of low oxygen.

2. The model for the role of hypoxia in lipid remobilization will benefit from further rigorous testing as the authors only use Lsd1-IR as the only manipulation for this.

To address this specific concern, we tested the effects of fat body-specific RNAi knockdown of two additional genes required for lipid synthesis – ACC (an enzyme which is required for the initial step of converting acetyl-coA to lipids) and lipin (which is involved in a later step in lipid synthesis). In both cases we observed phenotypes similar to Lsd2 RNAi – RNAi knockdown of lipin and ACC lead to reduced survival in hypoxic animals. These data are presented in Fig 8 c, d. These data strengthen our model that increased lipid synthesis in the fat body is required for organismal adaptation to hypoxia.

a. Figure 8: Also authors do not use appropriate controls for this, using $r4>+$ is not ideal as just expression of dsRNA in animals has physiological effects. Authors should use RNAi of a heterologous gene such as GFP-RNAi, LacZ or mCherry-RNAi as control.

As suggested by the reviewer, we have now included an experiment in which we test the effects of expression of GFP RNAi as a control. We found that expression of dsRNA to GFP had no effect on hypoxia survival. These data are presented in Suppl Fig 9d.

b. While the authors have discussed Brummer lipase, they have not performed the crucial experiment of using UAS-bmm (over-expression) or UAS-bmm-RNAi. The prediction of their model would be that inhibition of this lipase should provide hypoxia resistance whereas over-expression of bmm should phenocopy Lds2-IR.

We carried out these experiments. As predicted by the reviewer we found that brummer overexpression phenocopied Lsd2 IR – we saw that brummer blocked the increase in lipid droplet size seen in hypoxic larval fat bodies (Fig 8a) and that brummer overexpression reduced survival in hypoxic animals (Fig 8f). In terms of the potential protective effect of

brummer knockdown, as discussed in the response to point 1c above, the challenge with looking for protective effects is that larvae already show a high degree of hypoxia tolerance (~80-90% survival) and so it was difficult to observe any further increase in survival.

c. While the authors show that hypoxia phenocopies the effect of high sugar diet on Lipid droplets, their claims and richness of the model will be significantly enhanced if they can test whether larvae which have been exposed to a high sugar diet or larvae of naturally occurring fat mutants (such as adipose60) survive hypoxia better.

On the whole the experiments presented seem rather unidirectional (either over-expression or knock down) and I would like to see the authors make improvements along these lines. For instance, experiments done in Figure 7 are more convincing and exhibit a depth lacking in Figure 3,4 and 8 from a genetic stand-point.

As discussed above, since the animals on our normal lab diet (yeast/cornmeal/sugar) already show ~80-90% survival in hypoxia, it's difficult to see if any manipulations (e.g. high sugar diet, mutants) can lead to even better hypoxia survival. We did try an additional expt in which we examined the effects of a yeast-only diet and compared this to a yeast+glucose diet (see attached Fig1 and 2). We found that in normoxic conditions, addition of glucose to yeast diet lead to increase lipid droplet size in larval fat body cells and this increase mimicked the effects of hypoxia. However, in hypoxia, animals in both the yeast-only and yeast+glucose diet showed an increase in lipid droplet size (Fig 1) and larvae raised on both diets showed high levels of hypoxic survival (comparable to survival levels on our lab diet)(Fig2). These data suggest that just a minimal yeast-only diet has sufficient amounts nutrients (sugar, lipids) to support the hypoxia-induced increase in larval lipid stores and to allow for high levels of hypoxic survival. We found that lowering levels of yeast (below 2%) or further increasing levels of dietary glucose (above 1%) lead to larval growth delays and reduced development in normoxic conditions making it hard to interpret any effects in hypoxia.

We decided not to include this data in the paper, since they were not too informative. However, they do suggest future experiments looking at how hypoxia might increase lipid stores (e.g. via dietary sugars? via dietary lipids? - perhaps studying holidic defined diets would be the right approach). Also, it's unclear whether mobilization of glucose from fat body glycogen or directly from dietary glucose is the main starting point for new lipid synthesis. Experiments tracking C¹³-glucose metabolic flux in hypoxia larvae would be useful in the future. These experiments are all beyond the scope of the current work but we do now discuss these points in the paper discussion.

3. The data presented in Supplementary Figure 1, based on which the authors claim that hypoxia does not affect feeding is highly unsatisfactory and very subjective. The authors must present more convincing evidence to stake the claim that hypoxia does not affect food intake, else they need to leave this open-ended and inconclusive.

As suggested by the reviewer we have now carried out an additional experiment in which we measured mouth hook contractions as an index of feeding (a commonly used feeding behavior assay in larvae). We find that hypoxia has no effect on mouth hook contractions. These data are now presented in Supplemental Figure1b.

4. The sentences in results section whether authors use UAS-Rheb to ectopically activate TOR are written in quite a confusing manner. I would encourage the authors to edit this section to make it more

readable, clear and logical. I had to read it multiple times to follow the argument made by the authors. Page 6 (lines: 14 onwards).

We have now re-written this section to make it clearer.

Reviewer #2 (Remarks to the Author):

We thank the reviewer for their time and their thoughts about how to improve our paper. We've outlined our responses to their comments below:

Generally, the text is clear, figures are well presented and data convincing. The data would, in some cases however, be strengthened by combining data from repeat experiments, so that statistical significance can be established, eg in Figure 2 have the westerns been repeated and quantified? In

Figures 5, 7, 8a, it is not clear how many larvae were studied and if possible quantification of differences should be added.

We have now addressed each of these specific points as follows:

- 1) Two key western blot results from Fig2 are those showing that Rheb overexpression and *tsc1* mutation can prevent hypoxia-mediated decreases in TORC1 signaling. We had originally carried out these experiments in at three or four independent replicates. We have now quantified these results (using Image J) and performed statistical analysis, and they are presented in Suppl Fig3a, b. These quantifications show that hypoxia significantly represses TORC1 activity and that this effect is lost upon Rheb overexpression or loss of TSC. We have also added an additional experiment looking at effects hypoxia on TORC1 signaling in larvae transheterozygous for two independent *tsc1* mutant alleles (Suppl Fig 3c). These results also show that loss of *tsc1* completely prevents the hypoxia-mediated suppression of TORC1.
- 2) The other western blot experiments were originally performed with biological independent duplicates. We have now presented the duplicate experiments in Suppl Fig 5-7, and we have quantified these duplicate blots.
- 3) We have also performed additional experiments (also as independent biological duplicates) looking at the effects of *Sima* RNAi (to complement our *sima* mutant data) and *scylla*, *charybdis* double mutants (to complement the single mutant analysis and to also rule out genetic redundancy). For the *scylla* and *charybdis* expts, we quantified the double mutant expt (Suppl Fig6c,d) since this experiment provided the most rigorous test of the role for these two genes. In all these cases, the phospho-S6K western blots show that loss of *sima*, *scylla*, *charybdis*, or *scylla/charybdis* together, does not prevent the hypoxia-mediated suppression of TORC1 activity.
- 4) For figures 7 and 8, quantifications of lipid droplet sizes are now shown in Suppl Fig10a, b. Each of these experiments was performed by analyzing fat bodies from at least 10 different larvae per condition. Details about the quantification and replication are now included in the methods section.
- 5) For the LysoTracker expts in Fig 5, fat bodies from at least fifteen larvae per condition were analyzed. In all cases, we saw essentially no LysoTracker stained punctae in the normoxia and hypoxia conditions, but extensive LysoTracker positive staining in all fat bodies from the starved larvae. These details are now included in the methods, results and figure legends.

Some of the authors claims may be overstated, for example in Figure 2 – the authors claim that the control of mTORC1 activity by hypoxia is via TSC/Rheb. However, they do this by showing loss of TSC activity or overexpression of Rheb suppresses the phenotype. This does not exclude the possibility of a parallel non-dominant pathway being involved. They need to undertake molecular/biochemical studies to show this link exists.

As the reviewer suggests, our data doesn't rule out a non-dominant parallel pathway that may act in parallel to the TSC-Rheb pathway to control TORC1. We have now modified the interpretation of these data in the results and discussion section to reflect this point. The reviewer indicated that we '*need to undertake molecular/biochemical studies to show this link exists*'. We interpreted this to mean the TSC-Rheb-TOR link (rather than the putative non-dominant, parallel pathway link). The experiments we have carried out have used phosphorylation of S6K as the main molecular measure of TORC1 activity – this is perhaps the most widely used molecular assay for TORC1 activity in the field (both *Drosophila* and

mammalian cell culture). Our data showed that loss of TSC prevented hypoxia from suppressing TORC1. Other approaches to look at TSC-Rheb regulation (such as GTP-loading of Rheb or measuring phosphorylation status of TSC1/2) are very difficult to perform in larvae since they require [³²P]radio-labelling of cells and good antibodies (which we don't have). As our paper stands, we feel our genetic TSC expts combined with measuring phospho-S6K provide good support for stating that *“the rapid suppression of TORC1 signalling upon hypoxia exposure in larvae requires TSC1/2 function”* (page 6, lines 23-25).

A potential non-dominant parallel pathway would be difficult to identify (based on its non-dominant nature). We have provided two new pieces of testing two additional signaling molecules could have potentially regulated TORC1 – the ptp61F (a phosphatase previously reported to control TORC1 in Drosophila cell culture in responses to hypoxia – Lee et al MBC, 2008, 19,4051) and nitric oxide (which was previously shown to be induced by hypoxia in larvae and which was shown to be required for hypoxia tolerance - Teodoro & O'Farrell, 2003, EMBO J, 22, 580; Wingrove & O'Farrell, 1999, cell, 98, 105). However, our data suggest that neither seem to be involved. These data are presented in Suppl Fig 7b, c.

Minor points and typographical errors

Supp Figure 2b seems to show similar data to Figure 2b – could this data be combined?

We included Suppl Fig 2b (now Suppl Fig 2c) since this expt looked in more detail than the expt in Fig 2b at oxygen concentrations between 1 and 10%. Since the two blots each include different oxygen concentration conditions, we have kept them separate. However, we have included quantification of several independent experiments where we looked at the effects of 20 (normoxia), 10, 5, 3 and 1% oxygen (as in Fig 2b). This quantified data is shown in Suppl Fig 2b.

Page 18 (13 lines from the bottom): suggest changing ‘hypoxia larvae’ to ‘hypoxic larvae’

We have made this change

Page 19: Figure 4 is mislabelled as Figure 3

We have made this correction

Reviewer #3 (Remarks to the Author):

We thank the reviewer for their time and their thoughts about how to improve our paper. We've outlined our responses to their comments below:

1) Scylla and Charybdis have been proposed to be partially redundant, so the anti S6K western blots of hypoxic larvae extracts should be repeated in a Scylla-Charybdis double mutant. Then we will know to what extent is TOR inhibition independent of these two genes.

As suggested by the reviewer, we have now examined *scylla*, *charybdis* double mutants (to complement the single mutant analysis and to also rule out genetic redundancy). We found

that, like the single mutants, loss of *scylla/charybdis* together does not prevent the hypoxia-mediated suppression of TORC1 activity. These data are presented in Suppl Fig 6.

2) Si ma homozygous mutant larvae have been reported to die in hypoxia (although they are fully viable in normoxia). This reviewer suspects that there might be a problem with the sima mutant line. Please double-check this line and demonstrate that it is truly sima loss-of-function (the authors might want to utilize a Sima dependent transcriptional reporter and/or perform an anti-Sima western blot). Even if TORC1 inhibition depends on Sima, this manuscript will still provide highly valuable data, so I would be happy to recommend its publication.

We do find that in our hands the homozygous *sima* mutants are viable as larvae in normoxia, but show complete lethality when raised in hypoxia. Our experiments looking at effects on TORC1 involved raising the larvae to third instar in normoxia and then examining effects on TORC1 signaling upon shorter term hypoxia. We have now included data showing that the *sima* mutants are lethal when raised in hypoxia (Suppl Fig 4b). We also carried out an immunostaining antibody with an anti-*sima* antibody. This showed loss of nuclear *sima* staining in the *sima* mutants (Suppl Fig4a). In our hands, the antibody was very poor for western blots (many, many bands) and it was hard to discern the actual *sima* band.

To complement our *sima* mutant data, we have also included new data showing that ubiquitous *sima* knockdown using a *sima* RNAi also failed to reverse the hypoxia-mediated suppression of TORC1 signalling (suppl fig5b). Like *sima* mutants, we found that ubiquitous *sima* knockdown lead to complete lethality in larvae raised in hypoxia (Suppl Fig 4c) We also saw that *sima* RNAi showed a robust decrease in *sima* mRNA levels and a *sima* target gene, *fatiga* (Suppl Fig 4d)

Minor point: Please note that Figure 4 is mislabeled (Fig.3 instead of Fig 4).

We have now corrected this point.

REVIEWERS' COMMENTS:

Reviewer #1 (Remarks to the Author):

The authors have addressed my concerns and performed additional experiments which improve upon the original version. In this work, the authors here have identified a developmental window in a highly genetically tractable system in which the biology of hypoxia response can be probed further. This is a nice and rigorous study and I strongly support the publication of this work in Nature Communications.

Reviewer #2 (Remarks to the Author):

Overall I am happy with the response of the authors to my comments and those of other authors. My preference would be to place the quantification of the westerns for Figure 2 in that figure rather than in the Supplementary data.

Reviewer #3 (Remarks to the Author):

All my concerns have been properly addressed so I can recommend publication of the manuscript.

We thank the reviewers for taking the time to review our paper again.
We've included our responses (in blue) below

REVIEWERS' COMMENTS:

Reviewer #1 (Remarks to the Author):

The authors have addressed my concerns and performed additional experiments which improve upon the original version. In this work, the authors here have identified a developmental window in a highly genetically tractable system in which the biology of hypoxia response can be probed further. This is a nice and rigorous study and I strongly support the publication of this work in Nature Communications.

Thanks! You rock.

Reviewer #2 (Remarks to the Author):

Overall I am happy with the response of the authors to my comments and those of other authors. My preference would be to place the quantification of the westerns for Figure 2 in that figure rather than in the Supplementary data.

Thanks! You rock. We've made the change to Fig 2 as requested.

Reviewer #3 (Remarks to the Author):

All my concerns have been properly addressed so I can recommend publication of the manuscript.

Thanks! You rock.